# Mass Transfer Resistance and Reaction Rate Kinetics for Carbohydrate Digestion with Cell Wall Degradation by Cellulase

**DOI:** 10.3390/foods13182881

**Published:** 2024-09-11

**Authors:** Yongmei Sun, Shu Cheng, Jingying Cheng, Timothy A. G. Langrish

**Affiliations:** Drying and Process Technology Group, School of Chemical and Biomolecular Engineering, Building J01, The University of Sydney, Camperdown, NSW 2006, Australia; ysun2550@uni.sydney.edu.au (Y.S.); sche2348@uni.sydney.edu.au (S.C.); jche0791@uni.sydney.edu.au (J.C.)

**Keywords:** in vitro digestion, mass transfer, plant-based food digestion, starch hydrolysis, cellulase, plant cell wall

## Abstract

This paper introduces an enzymatic approach to estimate internal mass-transfer resistances during food digestion studies. Cellulase has been used to degrade starch cell walls (where cellulose is a significant component) and reduce the internal mass-transfer resistance, so that the starch granules are released and hydrolysed by amylase, increasing the starch hydrolysis rates, as a technique for measuring the internal mass-transfer resistance of cell walls. The estimated internal mass-transfer resistances for granular starch hydrolysis in a beaker and stirrer system for simulating the food digestion range from 2.2 × 10^7^ m^−1^ s at a stirrer speed of 100 rpm to 6.6 × 10^7^ m^−1^ s at 200 rpm. The reaction rate constants for cellulase-treated starch are about three to eight times as great as those for starch powder. The beaker and stirrer system provides an in vitro model to quantitatively understand external mass-transfer resistance and compare mass-transfer and reaction rate kinetics in starch hydrolysis during food digestion. Particle size analysis indicates that starch cell wall degradation reduces starch granule adhesion (compared with soaked starch samples), though the primary particle sizes are similar, and increases the interfacial surface area, reducing internal mass-transfer resistance and overall mass-transfer resistance. Dimensional analysis (such as the Damköhler numbers, *Da*, 0.3–0.5) from this in vitro system shows that mass-transfer rates are greater than reaction rates. At the same time, SEM (scanning electron microscopy) images of starch particles indicate significant morphology changes due to the cell wall degradation.

## 1. Introduction

Plant-based food production (fruits and vegetables, whole grains, beans, peas, nuts, and lentils) usually uses less energy, water, and land than animal-based foods [1,2]. Plant-based diets provide a sustainable solution to reduce greenhouse gas emissions and to address climate change [3]. Plant-based food digestion involves several mass-transfer steps during nutrient transport from a food to the digestive juice, through the gastrointestinal wall, to the blood stream, which form a sequence of film resistances to mass transfer [4]. Various nutrients (as food components) dissolve or diffuse at different mass-transfer rates. The overall equation for the mass-transfer rate of any food component is [5]:(1)$$N_{A}=K\;A\;\left(C_{1}-C_{2}\right)$$
where *N_A_* is the mass-transfer rate (kg·s^−1^), *K* is the overall mass-transfer coefficient (m·s^−1^), *A* is the interfacial surface area (between phases, m^2^), and *C*_1_ and *C*_2_ are the concentrations inside the food and inside the digestive juices or outside the food, respectively (kg·m^−3^).

For example, carbohydrate digestion involves a solid phase (starch granules) and a liquid phase (digestive juice), so the mass-transfer resistance layer includes the plant cell wall and other regions inside the plant cell (where cellulose is a significant component) [6]. The nature of the cell wall is a major difference between plant and animal cells [7,8]. Pioneering research has studied the internal mass-transfer coefficient or permeability constant of cell walls (membranes) from different plants [9,10]. For plant-based foods, starch granules are trapped in the plant cell walls (cellulose, hemicellulose, and pectin) [11,12], which is the internal mass-transfer resistance for starch hydrolysis. Enzymatic plant cell wall degradation has been used for industrial applications, such as biofuel and biogas production, and paper production, in recent decades [13,14]. Though “To date, the enzymatic conversion of cell-wall materials into fermentable sugars is characterized by low-efficiency and high operating costs”, chemical treatment is still broadly used to degrade plant cell walls [13], and this enzymatic approach provides an alternative way to understand the internal mass-transfer resistance from plant cell walls quantitatively, which is one example of a research gap for food digestion studies.

Mass transfer theory has also been employed in other research to understand starch hydrolysis (carbohydrate digestion). For instance, Santamaria et al. [15] mentioned that “This kinetics constant value could be interpreted like the kinetics constant in absence of mass transfer resistances with gel” and “The mass transfer coefficients value strictly depends on the characteristics of compound diffusing, turbulence conditions on the surface and properties of the fluid”, but there are limited quantitative data measuring internal and external mass-transfer resistances, component diffusivity, and relevant reaction rates or kinetics constants. In the research by Santamaria et al. [15]), the starch hydrolysis test was most likely run under a stirrer speed of 150 rpm, as it refers to the digestion study from Aleixandre et al. [16]. However, the mass-transfer coefficient and/or resistance estimated by Equation (7), involving a reaction kinetic constant, seems inappropriate. The equation ($\frac{1}{K}=\frac{1}{K_{1}}+\frac{1}{K_{m}}$) from Santamaria et al. [15] has been mentioned as below, “… a global kinetics coefficient where enzymatic reaction constant value (*k*_1_, min^−1^) and mass transfer coefficient (*k*_m_, min^−1^) are involved and the simplified relationship, after several assumptions for a model of resistances in series, …”. This equation is most likely from Chapter 25 (Equation (35)) in the book *Chemical Reaction Engineering* [17]. Equation (35) has indicated the mass-transfer resistances during combustion, and the enzymatic-reaction constant is not a form of mass-transfer coefficient in Equation (7) from Santamaria et al. [15].

An appropriate approach may be the use of a dimensionless group, the Damköhler number (*Da*), to assess the relative importance of mass transfer and chemical reaction in starch hydrolysis studies. The mass-transfer rate of a digestive enzyme may affect the reaction rate constant, but the two rates and time constants are defined separately, and a dimensionless group, the *Da* number, can connect these two rates and constants together. The *Da* is defined as the ratio of the time constant for overall mass transfer to the time constant for the reaction [18], allowing an assessment of the relative importance of mass transfer and chemical reaction in determining the overall reaction rate and time.

In this research, a proposal has been developed (Figure 1) to assess the internal mass-transfer resistance or plant cell wall permeability constant of starch (e.g., potato starch) by using cellulase to degrade the cellulose in the cell walls. Cellulose is the predominant component in plant cell walls [11,12]. In sample B, raw starch is treated or hydrolysed by cellulase to remove or reduce the cell walls, so its internal mass-transfer resistance will be reduced. The overall mass-transfer resistance in sample B, 1/*K*_B_ is equal to external mass-transfer resistance, 1/*K*_Bext_. For the same stirrer speeds in samples A and B, the external mass-transfer resistance in sample A is the same as that in sample B. If the overall mass-transfer resistance is estimated for sample A, 1/*K*_Atot_, then the internal mass-transfer resistance in samples A is 1/*K*_Aint_ = 1/*K*_Atot_ − 1/*K*_B_. In this study, glucose levels have been monitored during starch hydrolysis in a beaker and stirrer system, and these dimensionless groups have been calculated by dimensional analysis based on the test results.

This paper estimates the mass-transfer resistance of starch cell walls, using the above proposed method in the starch hydrolysis reaction during food digestion. It also demonstrates evidence that the enzymatic plant cell wall degradation reduces starch granule adhesion and increases the interfacial surface area, as shown by particle size analysis, reducing the internal mass-transfer resistance and the overall mass-transfer resistance. The significance of this work has been to compare the reaction rate of starch hydrolysis by salivary enzymes with the mass-transfer rate of cellulase-treated starch and granular starch. This research has applied mass-transfer and reaction engineering theory in a quantitative study of starch hydrolysis, and a dimensionless group, the Damköhler number (*Da*), has been calculated based on glucose measurements from a beaker and stirrer system.

## 2. Materials and Methods

### 2.1. Materials

Cellulase from *Aspergillus niger* (EC Number: 232-734-4, powder, ≥0.3 units/mg solid, Sigma-Aldrich, Australia), starch from potato (soluble, ACS reagent, residue on ignition (Ash) ≤ 0.4%, Sigma-Aldrich, Bayswater, VIC, Australia), amylase from *Bacillus licheniformis* (alpha-amylase, lyophilized powder, 500–1500 units/mg protein, Sigma-Aldrich, Bayswater, VIC, Australia), and phosphate buffer (pH 6.9, 0.1 M) were used in this research.

### 2.2. Methods

#### 2.2.1. A Beaker and Stirrer System

This research was conducted in a beaker and stirrer system. A hot plate stirrer (Model PC-420D, Corning, Glendale, Arizon, Sugar Land, TX, USA) and a beaker of 150 mL with a diameter of 56.3 mm and a height of 80.4 mm were used. A stirrer speed of 100 rpm or 200 rpm was set with a magnetic stirrer (cylinder shape, 40 mm long, 8 mm diameter) in the buffer solution. A temperature of 37 °C was set to mimic the human body temperature, and a pH of 6.9 mimicked the food digestion condition for the starch hydrolysis study [19,20].

#### 2.2.2. Starch Cell Wall Degradation by Cellulase

At the start of the experiment, the buffer solution (phosphate buffer, pH of 6.9, 50 mL), and 25 mg cellulase [21] were added into the beaker, then the beaker was placed on the hot plate stirrer with the setting of 100 rpm and the temperature of 37 °C. Once the temperature of the solution system reached 37 °C, 0.5 g of starch powder was added into this system. After reaction times of one hour and four hours, the samples were dried in a laboratory drying oven (LabTec, Melbourne, Australia) at 80 °C or 50 °C for 24 h. This process was also applied to starch powder only when making soaked starch samples for particle size analysis.

#### 2.2.3. Starch Hydrolysis by Amylase

These cellulase-treated starch samples and soaked starch samples were then hydrolysed by amylase. A ratio of 2.4% *w*/*w* amylase to starch was used [22]. The buffer solution (50 mL) was added into the beaker, then the beaker was placed on the hot plate stirrer with a rotational speed setting of 100 rpm or 200 rpm and a temperature of 37 °C. Once the temperature of the solution system reached 37 °C, amylase was added into this system. The glucose contents were measured by a glucose meter (Accu-Chek Performa, Roche, Millers Point, Australia) at certain time intervals, such as 1–5 h. Each measurement involved at least two duplicates.

#### 2.2.4. SEM Analysis

The starch particle morphology was observed using SEM (scanning electron microscopy). Starch samples were coated by a Quorum-SC7620 Mini Sputter Coater (Quorum Technologies, Lewes, UK). Images of the coated samples were obtained via a Phenom-Prox SEM (Phenom-World, North Brabant, The Netherlands).

#### 2.2.5. Particle Size Measurement

The particle size distribution of the starch samples was analyzed by a Mastersizer 3000 (Malvern Instruments, Malvern, UK) with a universal feeding funnel and a 100% feed flow rate at 2.2 bar.

### 2.3. Data Analysis

For *n*th-order reaction kinetics, the *Da* is defined as shown in Equation (2):(2)$$Da=\frac{time\;constant\;for\;overall\;mass\;transfer}{time\;constant\;for\;reaction}=K_{r}C_{0}^{n-1}\tau$$
where *K_r_* is the reaction rate constant (s^−1^), C_0_ is the initial concentration, *n* is the reaction order, and *τ* is the mean residence time (s), if the mean residence time is characteristic of the mass-transfer rate in the solution. The starch hydrolysis in this beaker and stirrer system is a first-order reaction [23,24], and the *Da* number is the ratio of the time constant for overall mass transfer to the time constant for the reaction.

The time constant for the overall mass-transfer process of the amylase may be defined by the equation below [25],
(3)$$\tau =\frac{V}{KA}$$

Here, *V* is the volume of the solution (m^3^), which does not change significantly during the experiment and is the same for all of the experiments, K is the overall mass-transfer coefficient for the amylase, and *A* is the interfacial contact area (m^2^), which is the starch particle surface area in the beaker and stirrer system.

The overall mass-transfer coefficient, K, can be estimated from the slope of the glucose concentration–time curve (*dC*/*dt*), the saturation concentration (*C_sat_*) of the glucose in the solution [26], the volume of the solution (*V*), and the interfacial area for the starch and the solvent (*A*), as follows:(4)$$K=\frac{V}{A}\;\frac{dC}{dt}\frac{1}{Csat}$$
(5)$$\frac{1}{K}=\frac{1}{k_{int}\;}+\frac{\;H}{k_{ext}}$$

The actual glucose concentrations were collected from the starch hydrolysis reaction products in the beaker and stirrer system, while the time constant (*τ*) was calculated by fitting Equation (6) to the measured concentration time data using the least squared error (LSE) method, where Equation (6) is the expected response for a constant external mass-transfer coefficient [25].
(6)$$C=C_{max}\left[{1-e}^{\left(-\frac{t}{\tau }\right)}\right]$$

Here, *C_max_* is the equilibrium concentration of glucose.

## 3. Results and Discussion

### 3.1. Degradation of Starch Cell Walls

The initial microscopic analysis of cellulase-treated starch indicated a significant change in particle morphology. The starch powder used in this test had the appearance of spherical particles (Figure 2), but the particles of cellulase-treated starch were non-spherical from the SEM images in Figure 2. For cellulase-hydrolysed starch cell walls, glucose generated through this hydrolysis (cellulolysis), and starch granules entrapped in the cell walls were released into the buffer solution. At the beginning of the hydrolysis, cellulose from starch cell walls, starch granules, and a small amount of glucose appeared in the system. At the latter stage of the hydrolysis, most of the cellulose was hydrolysed into glucose. Due to the low glass transition temperature of glucose (less than 40 °C) [27], it may stick to starch granules and form non-spherical particles during the drying of this mixture in a standard laboratory drying oven (at 80 °C as an initial setting). The drying temperature was set at 50 °C later for particle size analysis.

It was also observed that some yellow colour appeared in the starch sample that had undergone one hour of cell wall degradation, but the starch sample after four hours of degradation was not yellow at all (Figure 3). It is likely that the yellowness came from the dried starch cell wall, where cellulose is the main composition. This cellulose yellowing was reported by Ahn et al. (2019) as well [28], and a diagram of the yellowing behaviour of cellulose was shown in the graphical abstract in that literature. With longer cell wall degradation, such as for four hours, most of the cellulose (the starch cell wall) was hydrolysed into glucose. The starch and glucose were white (as a powder) after drying (in an oven) at a temperature of 50 °C.

### 3.2. Starch Hydrolysis with Degraded Cell Walls

As indicated in Figure 1, one hypothesis is that the cellulase hydrolysed the plant cell walls of starch (cellulose) and progressively reduced the internal and overall mass-transfer resistances for starch hydrolysis, so that the starch hydrolysis rates tend to increase, compared with starch powder.

The experiment results in Figure 4 showed that the reaction rate increased (the time constant for the reaction decreased) for starch hydrolysis after the cell wall was degraded by cellulase, compared with starch powder alone. There were three duplicates of the glucose measurements, and the standard deviations were very low (0.01–0.16) for all measurements, so the mean data only were included in these results.

The effect of the stirrer speed on the reaction rates for starch granules [29] was not found for cellulase-treated starch samples. In this beaker and stirrer system, the stirrer speed affects the external mass-transfer coefficients for amylase diffusion into the starch and glucose diffusion away from the starch. For starch granules, increasing the stirrer speed increased the overall reaction time constant (Table 1). Amylase activity may be decreased with greater stirrer speeds, for longer reaction times [30,31]. As shown in Table 1, for the cellulase-treated starch samples, increasing the stirrer speed and the degradation time resulted in initially constant reaction rates, corresponding to constant slopes for the starch hydrolysis curves (which are related to the overall mass-transfer rates). These observations will be explained in the following sections through particle size analysis (Section 3.3) and the estimation of the *Da* numbers estimation (Section 3.4).

### 3.3. Particle Size Analysis

The primary particle sizes for both soaked starch and cellulase-treated starch are similar (Figure 5). However, the cellulase-treated starch samples tend to have more fine particles than agglomerated ones (Figure 6), and the specific surface area increases for the starch that has undergone cell wall degradation (Table 2). The increase in the specific surface area is significant for the starch sample after one hour of degradation, when starch granules were released from degraded starch cell walls at a relatively higher rate, compared with starch samples after one hour of soaking in the buffer solution. After four hours, the surface area for both starch samples reached around 120 m^2^/kg, and soaking and cellulase had almost the same effect on cell wall breakage, reducing the internal mass-transfer resistance for starch hydrolysis. The effect of particle size on starch hydrolysis has been discussed in earlier research [32], and it should be noted that both particle size and surface area are also relevant to mass-transfer behaviour.

Starch cell wall degradation (cellulose hydrolysis by cellulase) significantly increased the reaction rates for starch hydrolysis (by amylase), compared with untreated starch powder (Table 1). This effect may be explained by the reduction in the internal mass-transfer resistance for starch granules due to the degradation of the cell walls. The morphology analysis in Figure 7 shows that the surface damage of starch particles occurred for both soaked starch and cellulase-treated starch, compared with starch powder. It appears that the cellulase-treated starch samples had more fine particles, which is consistent with the results of the particle size analysis (Figure 6).

### 3.4. Estimation of the Damköhler Number for Starch Hydrolysis with Degraded Cell Walls

According to the hypothesis from Figure 1, after starch cell wall degradation by cellulase, the internal mass transfer resistance should be relatively low, and the overall mass-transfer resistance should be almost equal to the external mass-transfer resistance. For example, the overall mass-transfer coefficient for starch samples after one hour degradation was estimated to be 1.8 × 10^−7^ m/s by Equation (4), based on the slope of the glucose concentration–time curve (*dC*/*dt*) at 100 rpm. Therefore, the overall mass-transfer resistance for this degraded starch samples was 5.5 × 10^6^ s/m at 100 rpm, which was almost equal to external mass-transfer resistance in this beaker and stirrer system. For starch powder hydrolysis at 100 rpm, the overall mass-transfer resistance was 2.8 × 10^7^ s/m (Table 3), and the internal mass-transfer resistance was estimated as 2.2 × 10^7^ s/m (2.8 × 10^7^–5.5 × 10^6^) by Equation (5). This internal mass-transfer resistance for starch was consistent with the results of the permeability study of plants cell walls [9]. An example calculation has been put in the Appendix A to give more details of the calculation procedure.

The range of *Da* numbers in Table 4 was calculated based on Equation (2), where the *Da* number is the ratio of the time constant for overall mass transfer to the time constant for the reaction [18]. For cellulase-treated starch hydrolysis, at different stirrer speeds (100 rpm and 200 rpm), the time constants for both overall mass transfer and overall chemical reaction did not change significantly, while starch powder showed different trends as indicated in the previous discussion [29]. The *Da* numbers were around unity for both starch powder hydrolysis and cellulase-treated starch samples, indicating the reaction rates were comparable with the mass-transfer rates.

## 4. Conclusions

Cellulose hydrolysis provides an enzymatic approach to remove the plant cell wall from plant-based foods, as a technique for measuring the internal mass-transfer resistance of the cell walls. In this research, starch samples with cell wall degradation showed increased hydrolysis rates and mass-transfer rates, compared with starch powder. Cell wall degradation reduces the internal mass-transfer resistance for starch hydrolysis (by amylase) during food digestion. This enzymatic method is helpful to estimate the internal mass-transfer resistance in this in vitro system, which is consistent with the results from a previous study using a solvent to remove the plant cell walls.

In this beaker and stirrer system, changes in the internal mass-transfer resistances result in different overall mass-transfer resistances, while the external mass-transfer resistances remain the same at a constant stirrer speed.

The price of cellulase enzymes is very high and, as a result, the price of implementing this technique for food products could be very high. It might be possible to use cellulase enzymes for the management of vegetable waste in the food industry, decreasing the cost of waste disposal and reducing pollution in this industry.

## Figures and Tables

**Figure 1 foods-13-02881-f001:**
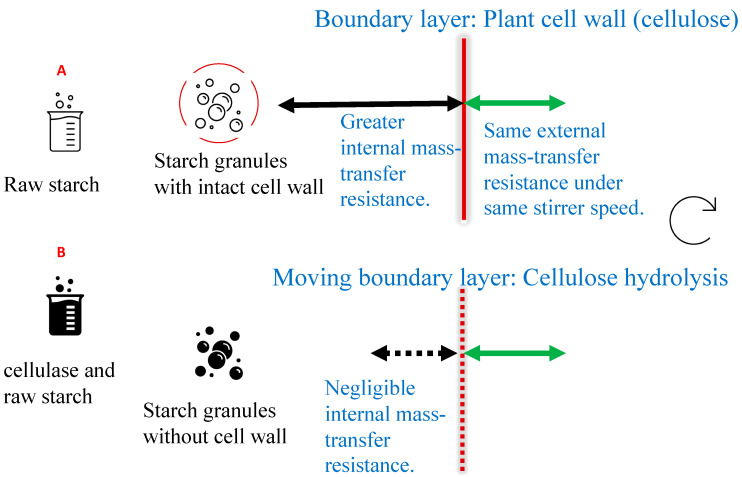
Diagram of mass-transfer resistance change for starch hydrolysis. (**A**) Raw starch; (**B**) Cellulase and raw starch.The plant cell wall affects the internal mass-transfer resistance. Degraded plant cell walls (cellulose) in sample (**B**) reduce the internal mass-transfer resistance.

**Figure 2 foods-13-02881-f002:**
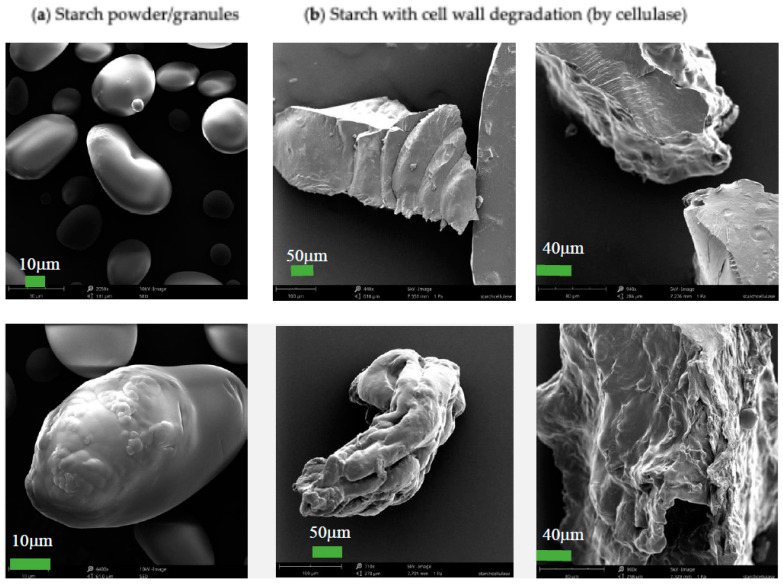
SEM images for the morphological analysis of (**a**) starch powder vs. (**b**) cellulase-treated starch (cellulolysis).

**Figure 3 foods-13-02881-f003:**
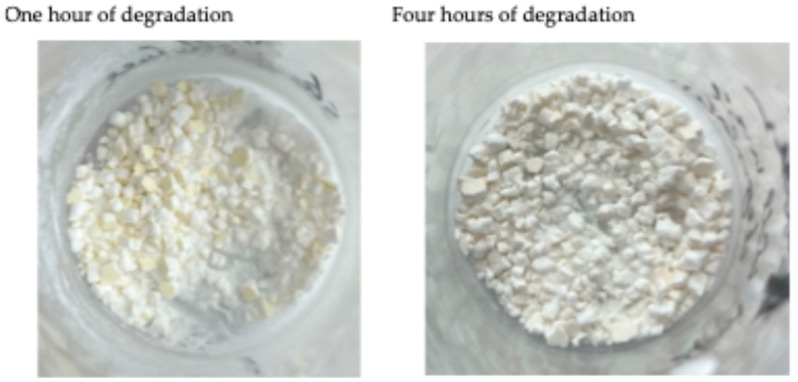
Evidence of starch cell wall degradation. Yellowish starch after one hour of cell wall degradation, but the yellow colour was almost gone after four hours of degradation.

**Figure 4 foods-13-02881-f004:**
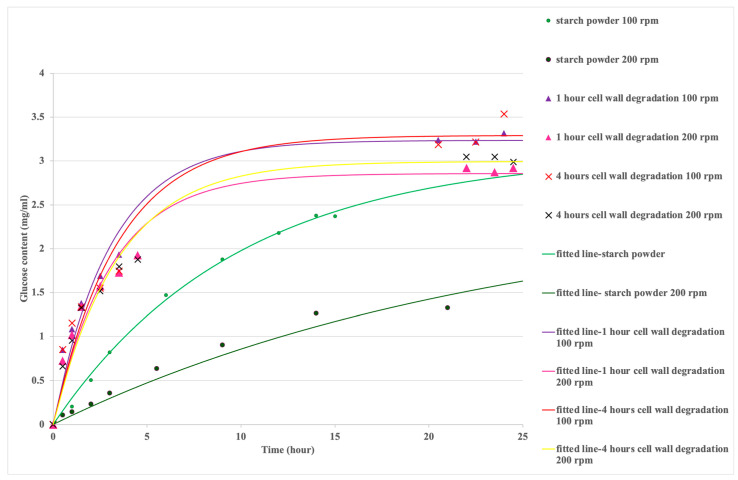
Starch hydrolysis before and after cell wall degradation (cellulose hydrolysis).

**Figure 5 foods-13-02881-f005:**
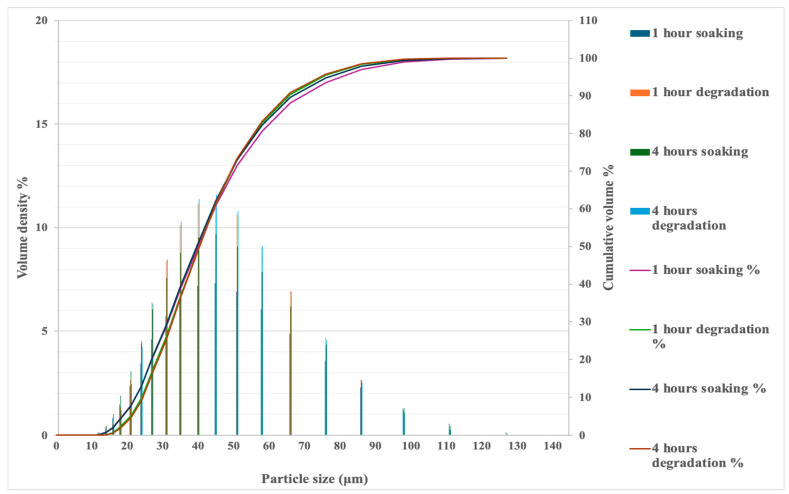
Cumulative particle size distributions for starch samples (soaking and degradation).

**Figure 6 foods-13-02881-f006:**
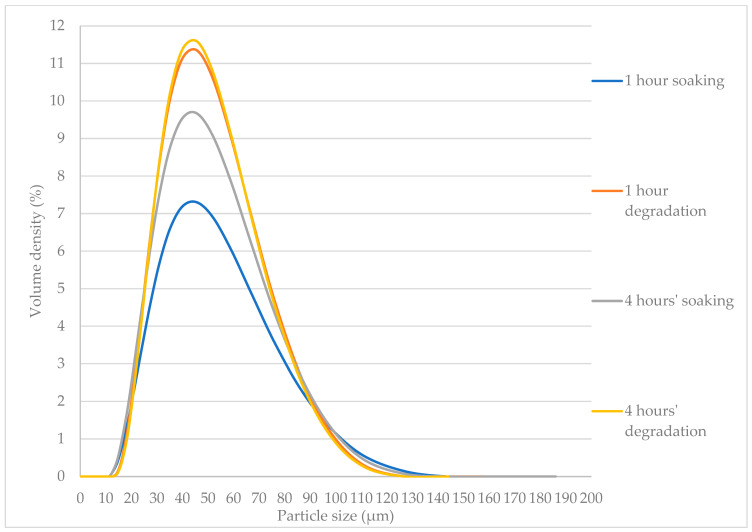
Particle size analysis for starch samples (soaking and degradation).

**Figure 7 foods-13-02881-f007:**
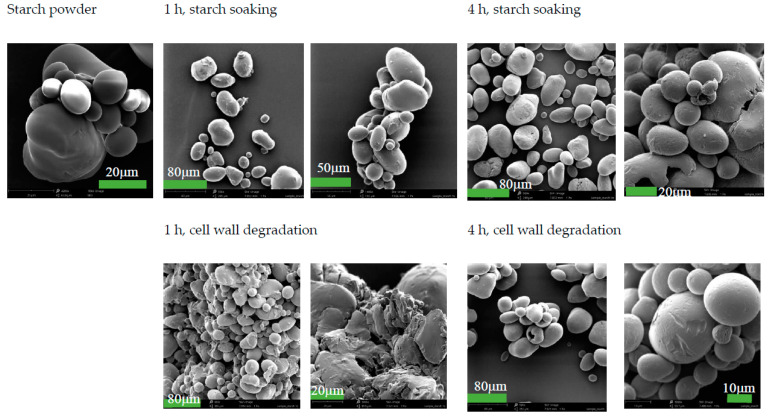
Morphology analysis of the changes in starch from soaking and cell-wall degradation. Samples were dried in a laboratory drying oven at 50 °C for 24 h.

**Table 1 foods-13-02881-t001:** Observed reaction time constants and slopes of the starch hydrolysis curve for starch powder and degraded starch samples (1% *w*/*w* starch in buffer solution and 2.4% *w*/*w* amylase/starch ratio).

Samples	Stirrer Speed (rpm)	Estimated Maximum Glucose Concentration (mg/mL)	Estimated Time Constant (tau) for the Reaction (h)	Slope of the Glucose Concentration–Time Curve (dC/dt) (mg/mL/s)
Starch powder	100	3.1	9.8	7.2 × 10^−5^
	200	2.5	24.1	2.8 × 10^−5^
1 h degradation—starch and cellulase	100	3.2	3.1	2.8 × 10^−4^
	200	2.9	3.1	2.7 × 10^−4^
4 h degradation—starch and cellulase	100	3.3	3.5	2.8 × 10^−4^
	200	3.0	3.5	2.6 × 10^−4^

**Table 2 foods-13-02881-t002:** Particle size analysis (Samples were prepared at 100 rpm, 37 °C, then dried in an oven at 50 °C overnight.).

Starch Samples	Specific Surface Area (m^2^/kg)
1 h, degradation	131 ± 0.4
1 h, soaking	96 ± 5
4 h, degradation	127 ± 3
4 h, soaking	118 ± 9

**Table 3 foods-13-02881-t003:** Estimates of mass-transfer coefficients and mass-transfer resistances for starch powder and cellulase-treated starch samples according to the hypothesis in Figure 1.

Samples	Stirrer Speed (rpm)	Estimated Overall Mass-Transfer Coefficient (m/s)	Estimated Overall Mass-Transfer Resistance (s/m)	Estimated Internal Mass-Transfer Resistance (s/m)
Starch powder	100	3.6 × 10^−8^	2.8 × 10^7^	2.2 × 10^7^
	200	1.4 × 10^−8^	7.2 × 10^7^	6.6 × 10^7^
1 h degradation—starch and cellulase	100	1.8 × 10^−7^	5.5 × 10^6^	Negligible
	200	1.7 × 10^−7^	5.8 × 10^6^	Negligible
4 h degradation—starch and cellulase	100	1.8 × 10^−7^	5.5 × 10^6^	Negligible
	200	1.7 × 10^−7^	6.0 × 10^6^	Negligible

**Table 4 foods-13-02881-t004:** The estimation of Damköhler numbers involved for starch powder and cellulase-treated starch samples by this in vitro system (1% *w*/*w* starch in buffer solution and 2.4% *w*/*w* amylase/starch ratio).

		Time Constant for Overall Mass Transfer (h)		Time Constant for Reaction (h)		*Da* (Time Constant for Overall Mass Transfer/Time Constant for Reaction)	
	Stirrer speed (rpm)	100	200	100	200	100	200
System							
Starch powder		4.6	12.0	9.8	24.1	0.47	0.50
1 h degradation (starch and cellulase)		1.2	1.2	3.1	3.1	0.38	0.40
4 h degradation (starch and cellulase)		1.2	1.3	3.5	3.5	0.34	0.36

## Data Availability

The original contributions presented in the study are included in the article, further inquiries can be directed to the corresponding author.

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
