# Peer review of "Mass Transfer Resistance and Reaction Rate Kinetics for Carbohydrate Digestion with Cell Wall Degradation by Cellulase"

_foods, 2024, doi:10.3390/foods13182881_

Round 1

Reviewer 1 Report

Comments and Suggestions for Authors

The present paper describes the mass transfer and kinetic studies of enzymatic hydrolysis of plant cell wall with a microbial celulase. The paper is interesting, however some revisions are required before it could be considered for publication, as follows:

1) Abstract – lines 9/10: The authors should clarify that the enzyme (celulase) was used to remove “plant cell walls” that contains high celulose concentration (pre-treament step) in order to improve the accessibility of amylases in starch hydrolysis. Initially, it seemed to me that cellulase was used in the hydrolysis of starch. This statement could confuse readers. I recommend revise all manuscript.

2) Introduction: The authors should provides an overview of recent developments in sustainable process design for enzymatic plant cell wall degradation. Is there a new design sustainable process and synthesis strategy than previously research? How is this system different to other reports to merit publication? Please, report.

3) The end of the introduction should be a remark of the interest of the study.

4) Materials and methods: Give technical information on the substrate centesional composition (celulose, hemicelulose and starch) and source (from potato¿¿) and enzymes (protein concentrration, units of activity and purity). These information are very importante to the readers.

5) Materials and methods: The authors should better describe the experimental apparatus to perform enzymatic hydrolysis, including initial units of enzyme activity used to perform this study, reaction volume, reactor dimension, temperature control method, stirrer type (mechanical or magnetic), intensity and impeller configuration. This is relevant to the readers.  I recommend a deep revision in this field.

6) The resolution of Figure 4 is poor. I recommend improve its quality.

7) All experimental results summarized in Fig. 4 and Tables 1–4 should be represented as mean ± SEM. Explain the number replications used in the study. Moreover, I recommend add cellulose hydrolysis percentage (and resulting starch composition) obtained for each set of experiments reported in Tables 1–4. Please, provide such experimental data.

Comments on the Quality of English Language

Minor editing of English language required.

Reviewer 2 Report

Comments and Suggestions for Authors

Introduction line 74-75 please explain what kind of plants did used in the study as raw material for starch. There are different types of starch from cereals, from potatoes, from tapioca etc. with different  characteristics.

Materials and methods line 93-95 please explain enzymes characteristics for cellulase, amylase (pH, maximum temperature, enzymatic activity etc). What kind of amylase did used in study alpha or beta amylase?

Line 101-103, "A temperature of 370C was to set to mimic the human body temperature" - the human body does not contain cellulase enzyme. 

Line 154 for the relevance of the study and apply in food technology please explain the source of starch, if is from natural source or industry; if is from potatoes, cereals, tapioca etc..

Line 169-175 please detailed and explain the process of yellow colour in the the starch after cellulase activity. Please refer to other studies from literature that obtain the same process like in to the current study. 

Line 199 amylase activity, please explain what kind of enzymes did used in the study alpha amylases or beta amylases, the amylase activity is total different. 

Line 281 Conclusions Please explain what is the cost in food technology to produce food products with cellulase? The price of cellulase enzymes is very high and as a result the price of food products would be very high. A propose to use of cellulase enzymes would be in the management vegetable waste in the food industry, decreasing the cost of waste disposal and reducing pollution in the food industry. 

Round 2

Reviewer 1 Report

Comments and Suggestions for Authors

The manuscript was revised, as suggested. Therefore, I recommend its accptance for publication.